# Differential Diagnosis and Management of Diarrhea in Patients with Neuroendocrine Tumors

**DOI:** 10.3390/jcm9082468

**Published:** 2020-08-01

**Authors:** Sara Pusceddu, Roberta Elisa Rossi, Martina Torchio, Natalie Prinzi, Monica Niger, Jorgelina Coppa, Luca Giacomelli, Rodolfo Sacco, Antonio Facciorusso, Francesca Corti, Alessandra Raimondi, Michele Prisciandaro, Elena Colombo, Teresa Beninato, Marta Del Vecchio, Massimo Milione, Maria Di Bartolomeo, Filippo de Braud

**Affiliations:** 1Department of Medical Oncology, Gastro-entero-pancreatic and Neuroendocrine Unit 1, ENETS Center of Excellence, Fondazione IRCCS Istituto Nazionale dei Tumori, Via Venezian 1, 20133 Milan, Italy; Martina.Torchio@istitutotumori.mi.it (M.T.); natalie.prinzi@istitutotumori.mi.it (N.P.); Monica.Niger@istitutotumori.mi.it (M.N.); francesca.corti@istitutotumori.mi.it (F.C.); Alessandra.Raimondi@istitutotumori.mi.it (A.R.); michele.prisciandaro@istitutotumori.mi.it (M.P.); Elena.Colombo@istitutotumori.mi.it (E.C.); teresa.beninato@istitutotumori.mi.it (T.B.); Maria.DiBartolomeo@istitutotumori.mi.it (M.D.B.); filippo.debraud@istitutotumori.mi.it (F.d.B.); 2Gastro-intestinal Surgery and Liver Transplantation Unit, Fondazione IRCCS Istituto Nazionale dei Tumori, 20133 Milan, Italy; robertaelisa.rossi@istitutotumori.mi.it (R.E.R.); Jorgelina.Coppa@istitutotumori.mi.it (J.C.); 3Department of Pathophysiology and Organ Transplant, Università degli Studi di Milano, 20122 Milan, Italy; 4Polistudium SRL, 20135 Milan, Italy; luca.giacomelli@polistudium.it; 5Department of Surgical Sciences and Integrated Diagnostics, University of Genoa, 16126 Genoa, Italy; 6Department of Medical Sciences, Section of Gastroenterology, University of Foggia, 71122 Foggia, Italy; saccorodolfo@hotmail.com (R.S.); antonio.facciorusso@virgilio.it (A.F.); 7Unit of Pharmacy, Fondazione IRCCS Istituto Nazionale dei Tumori, 20133 Milan, Italy; Marta.DelVecchio@istitutotumori.mi.it; 8Department of Pathology and Laboratory Medicine, Fondazione IRCCS–Istituto Nazionale dei Tumori, 20133 Milan, Italy; massimo.milione@istitutotumori.mi.it; 9Oncology and Hematology-Oncology Department, University of Milan, 20122 Milan, Italy

**Keywords:** neuroendocrine tumors, chronic diarrhea, exocrine pancreatic insufficiency, differential diagnosis

## Abstract

Diarrhea is a recurrent symptom in patients with neuroendocrine tumors (NETs) and can represent different etiologies; thus, differential diagnosis is challenging. This paper distinguishes the different causes of chronic diarrhea in patients with gastroenteropancreatic NETs, with the aim to identify the most appropriate therapeutic approach. Underlying causes of diarrhea can be multifactorial, including not only diarrhea that is related to specific hormonal hypersecretory syndromes, but also diarrhea that is secondary to the following: extensive surgery which can cause pancreatic exocrine insufficiency or short bowel syndrome, treatment with somatostatin analogs or other antineoplastic agents, and bile acid malabsorption. After initial management of diarrhea with general treatments (dietary modification, use of antidiarrheals), a proper differential diagnosis is necessary to treat patients with specific etiology-driven therapeutic approaches, such as somatostatin analogs, pancreatic enzyme replacement therapy, and tryptophan hydroxylase inhibitors. In conclusion, NETs should be considered in the differential diagnosis of patients suffering from chronic diarrhea, after the exclusion of more common etiologies. Furthermore, physicians should keep in mind that several different etiologies might be responsible for diarrhea occurrence in NET patients. A prompt diagnosis of the actual cause of diarrhea is necessary to guide the treatment and a multidisciplinary approach is mandatory.

## 1. Introduction

Neuroendocrine tumors (NETs) represent a heterogeneous group of tumors arising from enterochromaffin cells with characteristics of both nervous system and hormone-producing cells. NETs may occur in different sites, although the majority of them (70%) are disclosed in the gastrointestinal (GI) tract. Within the GI tract, the most common sites are the small bowel, pancreas, rectum, and colon [1].

NETs are rare tumors. However, data collected over the past 20 years in western countries showed an incidence of 1–5 per 100,000 inhabitants, with an increasing tendency due to improvements in identification and characterization [2,3]. The crude incidence of gastrointestinal NETs or gastroenteropancreatic (GEP) has markedly increased over the last years and is now estimated to be 5.25/100,000/year with a prevalence of 35/100,000/year [4].

Patients with GEP NETs can experience numerous and complex symptoms due to the hypersecretion of hormones and peptides such as serotonin, gastrin, insulin, glucagon [5], vasoactive intestinal peptide (VIP) [6], and pancreatic polypeptide [7], which can lead to specific hormonal hypersecretory syndromes. The most prevalent symptoms in patients with GEP NETs include diarrhea, fatigue, abdominal discomfort, flushing, and food intolerance [5].

Among them, diarrhea is a highly recurrent symptom in patients with NETs, but its actual incidence is likely underestimated. Underlying causes of diarrhea can be multifactorial, including not only diarrhea related to specific hormonal hypersecretory syndromes but also diarrhea secondary to the following: extensive surgery that can cause pancreatic exocrine insufficiency (PEI) or short bowel syndrome, treatment with somatostatin analogs (SSAs) or other antineoplastic agents, and bile acid malabsorption [5]. In clinical practice, diarrhea is often attributed to the presence of hormone-secreting tumors, even in cases of uncertain diagnosis. Indeed, the diagnosis of functioning NET should always depend not only on the presence of symptoms but also on the evaluation of specific biochemical markers. Therefore, it is important to keep in mind that other etiologies might be responsible for the occurrence of chronic diarrhea in NET patients.

This narrative review, based on the available literature and the authors’ experience, intends to help the physician distinguish among the different causes of diarrhea in patients with GEP NETs and to assist in identifying the most appropriate therapeutic approach.

## 2. Diarrhea: Basic Notions

Diarrhea may be defined in terms of stool frequency, consistency, volume or weight [8]. It is usually described as three or more loose or watery stools a day [9] (>200 g/day of stool with decreased consistency) [10]. Acute diarrhea typically resolves within 4 weeks (most commonly within 1 week) [9], while chronic diarrhea is defined as loose stools occurring more than three-times/day and lasting for more than 4 weeks [11]. It is estimated that 1–5% of adults suffer from chronic diarrhea [11,12,13,14,15].

### 2.1. Etiology Criteria

Diarrhea may have different etiologies; thus, the differential diagnosis is challenging [13]. Most common causes of acute diarrhea are infectious agents (such as rotavirus and Norwalk agent), drugs, and toxins [16] produced by bacteria [13], whilst chronic diarrhea is usually non-infectious. The pathophysiologic mechanisms underlying the occurrence of diarrhea include secretory, osmotic, inflammatory pathways and altered motility [17]. Other more specific causes can be found in patients with NET, e.g., the development of mesenteric fibrosis that can contribute to profound diarrhea by partial obstruction-knicking of the bowel and induce venous stasis, thus leading to malabsorption [18].

#### 2.1.1. Secretory Diarrhea

Secretory diarrhea is due to an altered electrolyte transport [17]. Secretory diarrhea can be determined by bacterial toxins, reduced absorptive surface area, luminal or circulating secretagogues (such as laxatives or bile acids), and clinical conditions that alter the regulation of intestinal function [13]. Secretory diarrhea is also associated with carcinoid syndrome (CS) due to the enhanced secretion of serotonin by tumor cells [16].

#### 2.1.2. Osmotic Diarrhea

Osmotic diarrhea occurs because of ingestion of poorly absorbable yet osmotically active solutes that prevent proper GI absorption of water and electrolytes [17]. An increased osmotic load can be measured in the stool [19] and, as a matter of fact, osmotic diarrhea might be investigated by assessing the fecal osmolar gap. However, fecal osmolarity measured directly in feces may not reflect the actual pathophysiology of diarrhea [19].

#### 2.1.3. Diarrhea Secondary to Altered Bowel Motility

Delayed intestinal transport in the small bowel, as in the case of blind loop syndromes or scleroderma, causes bowel stasis that promotes bacterial overgrowth and subsequent bile salt deconjugation. Therefore, enhanced colonic secretion and fat malabsorption determine diarrhea. Conversely, increased bowel motility can deliver excessively large volumes of stool to the colon. In this case, diarrhea occurs when the maximum colonic absorptive capacity (approximately 4 L a day) is exceeded [16].

#### 2.1.4. Inflammatory Diarrhea

In the case of inflammatory diarrhea, bloody stools are frequently observed and may be associated with fever, tenesmus or severe abdominal pain [17]. In general, this condition is due to the mucosa disruption and inflammation that are usually caused by ischemic colitis, idiopathic inflammatory bowel disease (namely ulcerative colitis or Crohn disease), and specific infectious processes. Moreover, neoplasia and radiation colitis may represent an alternative cause of inflammatory diarrhea. The presence of leukocytes or leukocyte proteins (e.g., lactoferrin or calprotectin) on stool examination is suggestive of inflammatory diarrhea. In addition, elevated C-reactive protein level or sedimentation rate and low serum albumin level may be helpful in the diagnosis of inflammatory diarrhea [17]. In the case of history or stool analysis suggesting chronic inflammatory diarrhea, colonoscopy or flexible sigmoidoscopy should be performed to evaluate for structural changes [17].

## 3. Diarrhea Associated with Peculiar Neuroendocrine Tumor Syndromes

### 3.1. Carcinoid Syndrome

Carcinoid syndrome (CS) is a paraneoplastic syndrome occurring in 30–40% of patients with well-differentiated GI NETs [20]. It is determined by endogenous secretion of peptide hormones (serotonin and kallikrein). CS is predominantly associated with NETs of the midgut with extensive liver metastases, since a large amount of tumor-secreted substances are not completely metabolized by hepatic cells and enter the systemic circulation, causing CS. The diagnosis of CS is clinical, based on the presence of symptoms of diarrhea (80% of the patients), hot flushes (50–85% of the patients) and wheezing (10–20% of the patients) (Table 1), and biochemical [21]. In detail, serotonin is metabolized into 5-hydroxyindoleacetic acid (5-HIAA), a biomarker measurable in the urine, which can be used to monitor treatment response in patients with CS [22]. High systemic serotonin levels are reflected by elevated urinary 5-HIAA [1]. The excess of serotonin increases peristalsis, reducing the absorption of water and electrolytes, and leading to diarrhea [1]. Twenty-four-hour urine collection for 5-HIAA has a sensitivity and specificity around 90% for small intestine NETs [21]. CS-related diarrhea is a chronic secretory watery diarrhea, with bowel movements ranging from 2 to 5 in a day to more than 20; it is not affected by fasting and is often associated with changes in hydro-electrolytic balance. Importantly, CS significantly and negatively affects patients’ quality of life (QoL) and results in changes in patients’ lifestyle, including diet, work, physical activity and social life [1].

### 3.2. Zollinger–Ellison Syndrome

Gastrinoma typically secretes high level of gastrin that causes acid hypersecretion. This condition is known as the Zollinger–Ellison Syndrome (ZES) [6], as it was described by Zollinger–Ellison in 1955 and is characterized by gastric and hydrochloric acid hypersecretion by the stomach’s oxyntic cells [16,23].

ZES is present in 18% of all GEP NETs [27]. In 90% of cases, these neoplasms are located in the head of the pancreas or in the second/third part of the duodenum. Gastrinomas are often multifocal with associated loco-regional lymph-node spread or liver metastases [28]. ZES is diagnosed when a very high level of gastrin is observed [6]; indeed, elevated basal gastrinemia levels are found in 98–99% of patients with this syndrome [16]. In particular, basal gastrinemia ten-fold higher than the maximum normal value is observed in about 40% of patients, and this finding is strongly suggestive of the presence of ZES. Patients with ZES are characterized by severe and recurrent peptic ulcer disease, pyrosis and chronic diarrhea (Table 1) [24]. In this case, the hypergastrinemia induces diarrhea by overwhelming the bowel with gastric secretions and acid inactivation of pancreatic enzymes [16]. Diarrhea is secretory and associated with steatorrhea, with a frequency of daily movements that can vary from 2 to 10 per day and is always associated with hypergastrinemia and acid hypersecretion. As a result, chronic diarrhea secondary to ZES benefits from proton pump inhibitor (PPI) therapy, which usually acts quickly and effectively [29].

### 3.3. Verner–Morrison Syndrome

Verner–Morrison syndrome, also known as VIPoma or watery diarrhea hypokalemia achlorhydria (WDHA) syndrome or pancreatic cholera, is due to hypersecretion of VIP. It is associated to rare NETs (incidence of 0.05–2.0%) that usually occur between the ages of 30–50 years and are mainly intrapancreatic (95%) [30]. Diagnosis of Verner–Morrison syndrome is made in presence of serum VIP levels of 250–500 pg/mL (it is necessary to measure VIP levels when the patient is symptomatic) and severe secretory diarrhea, usually >3.0 L per day [30], which can determine metabolic acidosis through bicarbonate depletion and hypokalemia [6]. Watery diarrhea related to an underlying VIPoma is not responsive to prolonged fasting (Table 1). Stool amount overcomes 700 mL per day and can exceed over 3000 mL per day in 70% of the cases [30]. VIP is responsible for the occurrence of this syndrome as it acts as a stimulator of electrolyte and water secretion from the intestinal tract, an inhibitor of gastric acid secretion, a promotor of blood flow, a vasodilator, and a regulator of smooth muscle activity [30]. Almost 9% of NETs associated with VIP hypersecretion arise in the presence of multiple endocrine neoplasia type 1 (MEN 1) syndrome [26].

### 3.4. Becker Syndrome (Glucagonoma)

Becker syndrome or glucagonoma is usually associated with the presence of large tumor (>5 cm) localized in the pancreatic body or tail in 80% of the cases [6]. Moreover, 50–80% of patients present advanced disease with metastatic liver lesions at first diagnosis [25]. Evaluation of the glucose and glucagon level is recommended for the diagnosis of glucagonoma as glucagon results inappropriately elevated (>500–1000 pg/mL) in this condition [24]. Becker syndrome is characterized by weight loss [25], anemia [6], typical skin lesions named necrolytic migratory erythema, diabetes mellitus or glucose intolerance, secretory diarrhea, mucosal abnormalities (cheilitis, glossitis and stomatitis), hypoaminoacidemia [25] and dyspepsia [24] (Table 1). In 5–17% of cases, this syndrome occurs in the setting of MEN-1 syndrome [26].

### 3.5. Syndrome Associated with Somatostatin Hypersecretion (Somatostatinoma)

Somatostatin hypersecretion syndrome is very rare, as only few hundred cases have been described in the literature [26]. The hypersecretion of somatostatin inhibits pancreatic exocrine and endocrine secretions, as well as the motility of the digestive tract. Therefore, diabetes mellitus, diarrhea/steatorrhea and cholelithiasis are usually observed (Table 1). Diabetes is due to the inhibitory action of somatostatin on insulin secretion and is generally mild and pharmacologically manageable [26]. Cholelithiasis is also a direct consequence of somatostatin inhibitory activity on the gallbladder motility [26]. Diarrhea and steatorrhea are caused by inhibition of the secretion of pancreatic enzymes and bicarbonate, impaired gallbladder and intestinal motility, as well as lower lipid absorption [26].

Tumors associated with this syndrome can be located at the pancreas or duodenum. At the time of diagnosis, liver and lymph node metastases are already present in the majority of the cases. The association with MEN-1 syndrome is rare, while they can more frequently arise in presence of type 1 neurofibromatosis [26].

## 4. Diarrhea Not Associated with Hormone Secretion (Non-Functioning Tumors)

The causes of diarrhea in non-functioning NET patients are numerous and include pancreatic exocrine insufficiency (PEI), bile acid malabsorption, short bowel syndrome, and side effects of antineoplastic treatments. Furthermore, in those cases where the etiology of chronic diarrhea in NET patients can be attributed neither to functioning forms nor to the most frequent causes of diarrhea in non-functioning NETs (as detailed above), other, mainly gastroenterological, etiologies should be taken into account.

Therefore, differential diagnosis is essential to avoid that non-functioning NETs or other causes of this symptom are mistakenly considered as functioning NETs due to the presence of diarrhea.

### 4.1. Diarrhea Caused by PEI

PEI is one of the most frequent causes of diarrhea in patients with well-differentiated, non-functioning NETs. PEI can be due to different factors, however, the iatrogenic effect induced by SSAs, the frequent pancreatic localization of the primary neoplasm, surgery of the GEP tract (pancreatic and upper GI surgery) and diabetes mellitus represent the main causes, especially in case of pancreatic NETs (Table 2) [31,32,33].

PEI is defined as inadequate availability of the pancreatic enzymes within the intestinal lumen [31,34,35]. Its onset is associated with the reduction in pancreatic enzymes below 10% of normal, and when minimal functional reserve has been reached.

Development of PEI can be due to different causes: (a) reduced production or secretion of pancreatic enzyme; (b) inadequate stimulation of pancreas secretion; (c) acid-mediated inactivation of pancreatic enzyme; and (d) obstruction of the pancreatic duct. In other terms, PEI could be considered as a multifaceted manifestation of multiple clinical conditions, which might initially produce fat maldigestion and malabsorption, finally resulting in malnutrition and weight loss [31,35]. PEI causes digestion to be impaired, which commonly manifests as diarrhea and steatorrhea [35,36], which is defined as an increase in fat excretion in the stools which tend to be pale, large volume, malodorous and loose [36].

In the case of steatorrhea, PEI should be suspected. Quantitative estimation of fecal fat (exceeding 7 g per 24 h) is an essential first step for the diagnosis of steatorrhea. For the evaluation of PEI, fecal elastase may be measured instead of the 72 h fecal fat testing [34]. A value <100 mcg/g is abnormal and indicative of PEI [36]. Furthermore, abdominal computed tomography (CT) or magnetic resonance imaging (MRI) scanning is recommended [36].

Surgical interventions are often associated with PEI when pancreatic and upper GI surgery, such as pancreaticoduodenectomy (DCP), distal pancreatectomy, gastrectomy, gastric bypass, is performed (Table 3).

PEI usually occurs when at least 90% of the pancreas has been resected. In particular, high incidence of PEI has been reported after DCP, most often performed to remove tumors from the head of the pancreas and with concomitant duodenal resection. In fact, pancreozymin, a fundamental enzyme for the secretion of pancreatic digestive enzymes, is a hormone secreted mostly by the duodenum cells and only to a lesser extent by the jejunum [37,38].

Non-surgical pancreatic pathologies causing PEI include unresectable pancreatic cancer, benign pancreatic tumor (before surgery), pancreatic diseases (including chronic pancreatitis), autoimmune pancreatitis, cystic fibrosis, Shwachman–Diamond syndrome, and sequelae of necrotizing pancreatitis. Among the non-surgical extra-pancreatic clinical conditions, type 1 and 2 diabetes, celiac disease and inflammatory bowel diseases, as well as other autoimmune disorders (as a result of disease activity or secondary to specific medications usually used in their treatment), such as Sjogren’s syndrome, HIV infection, tobacco use, elder age and drugs, should be considered in the differential diagnosis. PEI can be caused by several drugs, including SSAs [31,37,39].

In this specific setting, the pathogenic mechanism is based on the inhibition exerted by SSAs on the secretion of meal-stimulated digestive enzymes from the pancreas, particularly pancreozymin [1]. PEI is a recognized side effect of SSA treatment. It has been observed that one of the more common adverse reactions of chronic use of SSAs is steatorrhea, with frequencies of 26–65% for lanreotide, and 36–61% for octreotide [1], respectively. Stool collection for fat content analysis can be helpful for the diagnosis of steatorrhea associated with SSA treatment, but empiric therapy with pancreatic enzymes represents a more practical approach.

SSAs, especially given in high dose, can also cause a malabsorption syndrome as patients often describe foul-smelling, floating, roaming, and greasy diarrhea after meals. Diagnostic tests with fecal elastase-1 are reliable, especially in symptomatic patients, but are uncommonly used [1]. PEI, malabsorption and steatorrhea induced by SSAs are often responsive to treatment with pancreatic enzymes (i.e., CREON, Ultrase), it also being recommended by the North American Neuroendocrine Tumor Society (NANETS) guideline [22].

### 4.2. Diarrhea Caused by Bile Acid Malabsorption

Bile acid malabsorption may also cause diarrhea in patients with NETs, particularly in subjects with resection of the terminal ileum and/or right colon or cholecystectomy [1,21]. The symptoms of bile acid malabsorption display when bile salts reach the colon where they stimulate water secretion and increased intestinal motility, resulting in diarrhea [40]. The gold standard for diagnosing bile acid malabsorption is a selenium homotaurocholic acid conjugated with taurine (SeHCAT) scan, which presents a sensitivity of 100% and specificity of 89%, respectively. SeHCAT retention levels of 10–15% are considered to be mild, <10% as moderate and <5% as severe bile acid malabsorption [21]. However, these tests are not widely available in routine clinical practice [41] and, therefore, the diagnosis of diarrhea due to bile acid malabsorption is mainly based on history for specific surgery (i.e., resection of the terminal ileum and/or right colon or cholecystectomy) and after the exclusion of other etiologies of chronic diarrhea.

### 4.3. Diarrhea Secondary to Short Bowel Syndrome after Extensive Small Bowel Resections

Patients who have undergone extensive small bowel resections may suffer from diarrhea secondary to short bowel syndrome, although data on the prevalence of this syndrome in patients with GEP NETs are not available [21]. As this condition can lead to significant malabsorption and consequent weight loss, parenteral nutrition might be necessary in extreme cases [1]. Routine blood tests (complete blood count and metabolic profile) are required to evaluate fluid and electrolyte balance and nutritional status [13]. The onset of such diarrhea can usually be traced back to the time of surgery. In particular, resections of long (>100 cm) segments of the small bowel may lead to additional steatorrhea, making diagnosis and therapy difficult. It is important to consider that this form of diarrhea may be suspected when clinical course is not improved by high doses of octreotide [1].

### 4.4. Diarrhea Associated with Antineoplastic Treatments in NETs

Systemic chemotherapy and combination therapy with interferon, everolimus or tyrosine kinase inhibitors are known to exhibit additional GI side effects, including diarrhea (Table 4) [11]. Of note, diarrhea due to different medications, including antineoplastic agents, is often secretory, however an inflammatory component cannot be excluded and, as a matter of fact, some patients treated with chemotherapy recount bloody and/or exudative diarrhea depending mainly on iatrogenic mechanisms.

### 4.5. Other Causes of Diarrhea

The presence of alarm features, such as GI bleeding and/or anemia, fever, significant weight loss and/or nocturnal symptoms usually rule out the functional etiology of the diarrhea. In detail, the differential diagnosis should comprise IBS, celiac disease, inflammatory bowel disease (IBD), diarrhea related to other neoplasm, factitious diarrhea by taking laxatives, diarrhea after radiation and other drugs, and thyroid disfunctions.

In the case of watery diarrhea, celiac disease should be promptly ruled out by serology test [48]. In the case of positive test, an upper GI endoscopy with duodenal biopsies should be performed to confirm the diagnosis. In the case of inflammatory diarrhea, IBD should be suspected, a fecal calprotectin test together with complete blood tests including inflammatory markers are suggested [49]. in the cases of markedly elevated calprotectin levels consistent with high suspicion of underlying IBD (i.e., alarm signs, elevated inflammatory markers, pathological imaging tests, positive familiar history), a pancolonoscopy with terminal ileum intubation and multiple biopsies should be performed to confirm the IBD diagnosis [50,51]. Even in the cases of macroscopic normal features at colonoscopy, multiple biopsies should be performed in patients with chronic diarrhea to rule the diagnosis of microscopic colitis out [50,51,52].

Several endocrinopathies might lead to chronic diarrhea [13]. Hyperthyroidism and hypothyroidism can both be responsible for the occurrence of diarrhea through opposite pathogenic mechanisms [53]. In detail, up to 25% of subjects affected by hyperthyroidism present mild-to-moderate diarrhea with frequent bowel movements due to intestinal hypermotility, as well as a hypersecretory state of the intestinal mucosa. Conversely, in patients with hypothyroidism, diarrhea is mainly due to enhanced bacterial growth secondary to bowel hypomotility that characterizes this condition [53].

In elderly patients with alarm features including GI bleeding and/or weight loss, a colon neoplasia should be suspected and a pancolonoscopy should be the test of choice for the final diagnosis [13].

In all cases of unexplained chronic diarrhea, a pharmacological investigation should be always performed as many drugs can cause diarrhea [13] and the actual mechanism by which some drugs determine diarrhea is still unknown [54]. Magnesium-containing antacids, antimicrobials, lactose- or sorbitol-containing products, laxatives, nonsteroidal anti-inflammatory drugs, colchicine, prostaglandins, antineoplastics, cholinergic agents and antiarrhythmic drugs are the most commonly involved [55]. Some patients produce factitious diarrhea by taking laxatives; however, the diagnosis is not straightforward, and physicians should keep this possibility in mind [54]. It is also important to consider that radiation can determine chronic diarrhea that may occur years after exposure and this should be detected by thorough examination of a patient’s history.

## 5. Diagnostic Workup

In the case of a patient with a confirmed diagnosis of NET and who suffers from chronic diarrhea, the differential diagnosis should include: (i) functional syndrome, including carcinoid syndrome and other specific functioning syndromes; (ii) diarrhea secondary to surgery, namely PEI after pancreaticoduodenectomy and short bowel syndrome after extensive intestinal resection; (iii) diarrhea secondary to SSA treatment, namely PEI; (iv) diarrhea secondary to bile acid malabsorption; (v) diarrhea secondary to other antineoplastic treatments; and (vi) other gastroenterologic causes of diarrhea (Figure 1).

## 6. Management Options and Treatment Strategies

Although initial management can be conducted with preliminary steps, such as dietary modification, considered that chronic diarrhea in NET patients may be determined by several different pathogenic mechanisms, a proper differential diagnosis and the consequent prompt definition of specific etiology-driven therapeutic approaches is mandatory (Figure 2) [1].

### 6.1. General Treatments

#### 6.1.1. Dietary and Nutritional Intervention

Several nutritional issues should be considered in the management of chronic diarrhea. The recommended daily intake of water/fluids is at least 2 L per day and water and fluid intakes are crucial in such patients to prevent dehydration and revert electrolyte alterations such as hypokalemia. Indeed, potassium supplementation might be helpful. Alcoholic beverages, coffee and caffeine-containing drinks and carbonate soft drinks should be avoided or consumed sparingly in view of their amine content which might trigger diarrhea [11].

#### 6.1.2. Antidiarrheals

Anti-motility agents may be prescribed for symptomatic improvement of NET-related diarrhea in patients refractory to SSAs and who experience the GI side effects of SSAs [11]. Although therapy with nonspecific antidiarrheals (e.g., loperamide or diphenoxylate), may improve the diarrhea, evidence from prospective trials is lacking. Moreover, side effects may occur, especially with the use of anticholinergic drugs in the elderly. Finally, it is important to consider that antidiarrheals act as symptomatic drugs without addressing the underlying pathogenic mechanism, nor reducing circulating serotonin levels [1].

#### 6.1.3. 5-HT3 Receptor Antagonists, Antihistamine–Antiserotonin Compound

Small retrospective studies have highlighted that ondansetron has some activity in patients with CS-related diarrhea, but no prospective studies have compared it with other therapies. Although cyproheptadine has been reported to be effective in the management of CS diarrhea, it is rarely used for CS-related diarrhea given the availability of safer and more effective treatment options [1]. Cuban zeolite is known to be able to adsorb remarkable amounts of the biogenic amine histamine and water, therefore, a medical device containing zeolite (Detoxsan^®^ powder) has been reported to significantly improve symptoms of NET-related diarrhea [56]. Natural zeolites present attractive properties, such as ion-exchange, adsorption and molecular sieving [56], however, strong evidence is lacking.

### 6.2. Etiology-Driven Specific Treatments

#### 6.2.1. Diarrhea Associated with Carcinoid Syndrome or other Hormonal Hypersecretion Syndromes

Somatostatin Analogs (SSAs): Octreotide and lanreotide present a similar pharmacodynamic profile, binding with high affinity to somatostatin receptors 2 (SSTR2) and with moderate affinity to SSTR5. According to current guidelines, treatment with either octreotide or lanreotide is recommended as the initial first-line treatment in patients with CS (Table 5) [1]. If treatment with the standard dose is not effective, high-dose SSA therapy can be administered [57,58,59].

Tryptophan Hydroxylase Inhibitors: Telotristat, an oral inhibitor of tryptophan hydroxylase, has been investigated in TELESTAR and TELECAST phase III studies (Table 5) [72]. Telotristat ethyl improves CS diarrhea by intracellular inhibition of tryptophan hydroxylase and blocks serotonin production within NET cells [73]. Based on these studies, the US Food and Drug Administration (FDA) and the European Medical Agency (EMA) have approved telotristat for CS diarrhea [74].

Interferon-α: The interferons (IFNs) IFN-α2a and IFN-α2b bind to IFN receptors and activate a signal transduction cascade, leading to the transcription of multiple tumor suppressor genes in NET cells [22]. IFN indications are similar to those of SSAs, and this treatment can be effective in SSR-negative tumors [4]. Treatment with recombinant IFN-α2a or IFN-α2b has been associated with symptomatic remission (especially for flushing compared with diarrhea) in 30% to 70% of the patients affected by CS [22]. Symptom control has been reported to be similar in patients treated with IFN or SSAs, although the onset of response is delayed [22]. The most frequent adverse events associated with INF treatment are weight loss, anorexia, fatigue, and dose-dependent bone marrow toxicity, thus, assessment of complete blood and platelet counts at baseline, 1 and 2 weeks after starting treatment and monthly thereafter is required. Available data suggest a possible synergism between SSAs and IFN in CS treatment [22]. However, in real-life clinical practice, IFN is only rarely prescribed to treat chronic diarrhea due to its relevant side effects.

#### 6.2.2. Diarrhea Caused by Bile Acid Malabsorption

In confirmed cases or whenever there is a high suspicion of bile acid-induced diarrhea, an empiric trial of bile acid sequestrants, such as colestipol, cholestyramine, or colesevelam, might be reasonable, starting at a low dose and, thereafter, titrating to response [1].

#### 6.2.3. Pancreatic Enzyme Replacement Therapy in Patient with Pancreatic Insufficiency

Pancreatic enzyme replacement therapy (PERT) is the only available safe and effective treatment for PEI [75]. Roberts et al. in their review of patients undergoing DCP for periampullary malignancy, evaluated the association between PERT (Creon, Mylan pharmaceuticals) and overall survival. At a post hoc subgroup analysis, PERT (median dose 75,000 units) use was independently associated with improved survival on multivariable analysis (HR 0.72, 95% CI: 0.52–0.99; *p* = 0.044) and on propensity-matched analysis (*p* = 0.009). This effect of PERT upon improved survival was predominantly observed in patients with a dilated pancreatic duct (≥3 mm) [76]. The pilot study by Landers et al. suggested that PERT (starting dose 50,000 IU Creon per meal and 25,000IU for a snack) is a safe and potentially effective therapy for the treatment of PEI also in patients diagnosed with advanced pancreatic cancer, improving QoL as well [77]. The addition of pancreatic enzymes with meals appears to improve symptoms such as foul-smelling, floating, foaming, and greasy diarrhea after meals due to treatment with, especially high doses SSAs, in NET patients [22].

PERT (namely Creon) has been shown to significantly improve fat digestion and symptoms after pancreatic resection and in PEI [76], as reported also in randomized controlled trials which showed that replacement therapy improved fat absorption after 3 week trial periods [78,79,80], and to our knowledge, there are no data regarding the potential role of pancreatic enzymes in the neuroendocrine setting.

#### 6.2.4. Diarrhea Secondary to Short Bowel Syndrome after Extensive Small-Bowel Resections

In the acute phase, when metabolic imbalance with fluid leaks as well as gastric hypersecretion tend to occur, a close monitoring of the patient’s total output (both fecal and urinary) and prompt intravenous replacement of fluid and electrolyte losses is crucial [80]. Parenteral nutrition is the milestone of the treatment of short bowel syndrome and should be initiated as soon as the patient stabilizes after surgery. The adaptation phase is characterized by structural and functional changes to improve nutrient absorption and slow GI transit. During this phase, usually lasting 1–2 years, the patients should eat by mouth. There is no specific diet for individuals with short bowel syndrome, but patients should eat at least five or more small meals/day and avoid concentrated sugars; furthermore, vitamin or mineral supplementation might be necessary [81]. Approximately 50% of prolonged acute intestinal failure evolves to chronic intestinal failure (CIF) [82], which requires home-based parenteral nutrition (HPN) and various drugs, including common anti-diarrheal medication (e.g., loperamide, codeine), PERT, bile acid resins such as cholestyramine, antibiotics for bacterial overgrowth, lactase supplement, and drugs that reduce the frequency and volume of total parenteral nutrition (e.g., teduglutide) [81].

## 7. Conclusions

NETs should be considered in the differential diagnosis of patients suffering from chronic diarrhea, particularly after the exclusion of more common etiologies. Once the diagnosis of NET has been established, it is important to keep in mind that diarrhea is a very frequent symptom in patients with NETs either with or without CS, and its actual incidence is probably underestimated. Mechanisms underlying the occurrence of diarrhea in NET patients are multiple and often challenging to diagnose. The majority of physicians tend to, erroneously, attribute diarrhea always to CS, also in those cases where other etiologies or iatrogenic causes may be further responsible for symptom’s occurrence.

NET patients, even in the setting of advanced disease, are usually characterized by a long life expectancy, thus, the occurrence of chronic diarrhea as a direct effect of the tumor itself or as a consequence of various treatments, can be very troublesome for patients. Indeed, for patients with metastatic NETs, diarrhea remains a major clinical problem with high symptom burden resulting in reduced QoL and negative financial impact. Therefore, clinicians cannot underestimate this symptom as the proper management of chronic diarrhea not only improves QoL, but might also increase a patient’s adherence to medical treatments. In fact, occurrence of G3 diarrhea, if undertreated, is responsible for the dose reduction in several medical treatments, thus affecting their efficacy. As management can be complex, a multidisciplinary team set-up is mandatory for detailed investigations to allow early diagnosis and to improve and maintain a good QoL for patients [40]. It has been observed that several conditions can result in diarrhea and GI symptoms, therefore, objective evidence is necessary to make the correct diagnosis and to treat the patient with confidence [11]. In terms of therapeutic approach, although initial management can be conducted with preliminary steps such as dietary modification and anti-motility agents, etiology-driven specific treatments are required. In the case of functioning syndromes, SSA treatment is the gold standard. FDA and EMA has approved telotristat for diarrhea secondary to CS. PERT remains the only treatment for PEI [75] and it is safe and effective [34], as reported by randomized controlled trials. However, no data are available regarding the potential role of PERT in NET patients with a confirmed diagnosis of PEI. In confirmed cases of bile acid-induced diarrhea, an empiric trial of bile acid sequestrants, such as colestipol, cholestyramine, or colesevelam, might be reasonable. Finally, parenteral nutrition is the milestone in the treatment of short bowel syndrome and should promptly begin once the patient stabilizes after surgery.

In summary, chronic diarrhea is common in GEP NET patients and physicians should keep in mind that several etiologies, rather than CS alone, might be responsible for diarrhea occurrence. A prompt diagnosis of the right cause of diarrhea is necessary to guide the treatment and a multidisciplinary approach is mandatory. As data focused on this specific topic are scanty, we do hope that the current review might be helpful for clinicians in the differential diagnosis of chronic diarrhea in patients with GEP NETs, in order to identify the best therapeutic approach for each single patient. Further prospective studies are warranted to define standard treatment protocols in this setting.

## Figures and Tables

**Figure 1 jcm-09-02468-f001:**
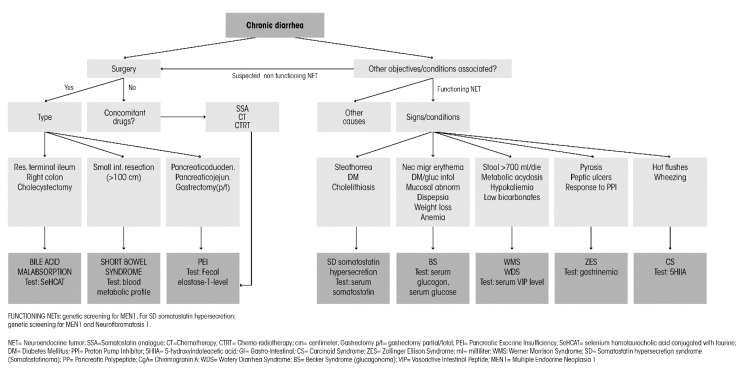
Differential diagnosis of chronic diarrhea in the case of a patient with a confirmed diagnosis of neuroendocrine tumor (NET).

**Figure 2 jcm-09-02468-f002:**
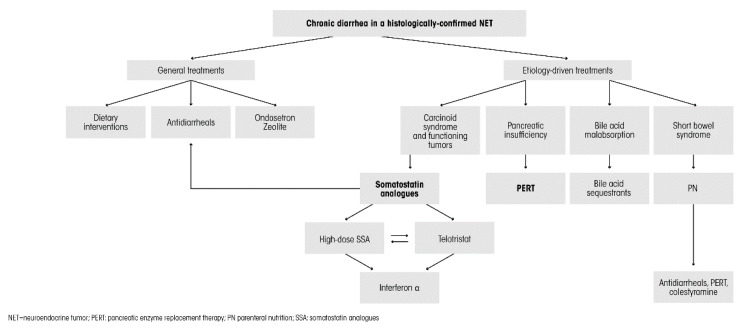
Algorithm for the treatment of diarrhea in NET patients, as applied in the authors’ institution.

**Table 1 jcm-09-02468-t001:** Signs and symptoms of the different hormonal hypersecretory syndromes.

Syndrome	Signs and Symptoms
Carcinoid syndrome	Diarrhea (mild or severe, usually after meals), hot flushes and wheezing, less frequently, heart failure, vomiting and bronchoconstriction [21]
Zollinger–Ellison syndrome	Severe peptic ulcer disease, pyrosis and recurrent diarrhea with watery or oily stools, responsive to PPI [23,24]
Verner–Morrison syndrome	Persistent watery diarrhea (after a 48 h fast) associated with depletion of liquids and electrolytes, stool amount overcomes 700 mL per day, metabolic acidosis through bicarbonate depletion, hypokalemia [6]
Becker syndrome (glucagonoma)	Weight loss [25], anemia [6], typical skin lesions named necrolytic migratory erythema, diabetes mellitus or glucose intolerance, secretory diarrhea, cheilitis, glossitis, stomatitis, hypoaminoacidemia [25] and dyspepsia [24]
Syndrome associated with somatostatin hypersecretion (somatostatinoma)	Diabetes mellitus, diarrhea, steatorrhea, and cholelithiasis [26]

PP: pancreatic polypeptide; PPI: proton pump inhibitors.

**Table 2 jcm-09-02468-t002:** Prevalence and factors associated with pancreatic exocrine insufficiency (PEI) in different clinical conditions.

Disease	PEI Prevalence (%)	Factors Associated with PEI Occurrence
**PEI Caused by Pancreatic Disorders**		
Chronic pancreatitis	30–90	Long disease duration Alcoholic etiology Extensive calcification Ductal obstruction
Acute pancreatitis	Mild: 15–20 Severe: 30–40	Necrosis extent (>30) Alcoholic etiology
Autoimmune pancreatitis	30–60	
Unresectable pancreatic cancer	20–60	Head localization Large site Ductal obstruction Coexistent chronic pancreatitis
Pancreatic neoplasm after surgery	Pancreaticoduodenectomy: 80–90 Distal pancreatectomy: 20–50	Whipple intervention Gastropancreatic anastomosis
Benign pancreatic tumor (before surgery)	30–60	Head localization Large size Ductal obstruction Coexistent chronic pancreatitis
Cystic fibrosis	80–90	Class I, II, III, IV *CTFR* mutations
Scwachman–Diamond syndrome	80–90	
**PEI Caused by Extrahepatic Disorders**		
Type I diabetes	30–50	High insulin requirement Poor glycemic control Early diabetes onset
Type II diabetes	20–30	Insulin requirement Poor glycemic control Long diabetes duration
Inflammatory bowel disease	Ulcerative colitis: 10 Crohn’s disease: 4	Disease reactivation (only for temporary PEI) Long disease duration Surgical patients
Celiac disease	5–90	Untreated disease (no gluten-free diet)
Pediatric intestinal transplantation	10	
HIV syndrome	10–50	Retroviral therapy
Gastrointestinal surgery	Total/subtotal gastrectomy: 40–90	Extensive intestinal resection
Esophagectomy	16	Vagal denervation
Sjogren’s syndrome	10–30	
Aging	15–30	Age > 80 years
Tobacco usage	10–20	Alcohol usage
Somatostatin analogs therapy	20	

PEI: pancreatic exocrine insufficiency; HIV: human immunodeficiency virus. Modified from [31].

**Table 3 jcm-09-02468-t003:** Pathogenic mechanisms involved in surgical conditions that lead to pancreatic exocrine insufficiency.

Surgery	Effect of Surgery			
	Decreased Neural Stimulation of Pancreatic Secretion	Decreased Hormonal Stimulation of Pancreatic Secretion	Postcibal Asynchrony	Loss of Pancreatic Tissue
Pancreaticoduodenectomy	Present	Present	Present	Present
Pancreaticojejunostomy	Present	Present	Present	Absent
Gastrectomy	Present	Present	Present	Absent
Partial gastrectomy	Present	Present	Present	Absent

PEI: pancreatic exocrine insufficiency. Modified from [33].

**Table 4 jcm-09-02468-t004:** Incidence of diarrhea as adverse event (AE) of medical treatments used in gastroenteropancreatic neuroendocrine tumors (GEP-NETs).

Therapeutic Drug	Percentage of Patients with Diarrhea as AE in the Different Arms	RCT	Patient Number	Reference
Everolimus (10 mg/day) + octreotide depot vs. placebo plus octreotide depot	All grades 27% vs. 16%	Randomized, Double-blind Placebo-controlled, Multicenter Phase 3 Study in Patients With Advanced Carcinoid Tumor (RADIANT-2) (NCT00412061)	429	[42]
Everolimus (RAD001 10 mg/day) + best supportive care vs. placebo + best supportive care	All grades 34% vs. 10%	Randomized Double-blind Phase 3 Study in Patients With Advanced NET (RADIANT-3) (NCT00510068)	410	[43]
Everolimus (RAD001) + best supportive care vs. placebo + best supportive care	All grades 31% vs. 16%	Randomized, Double-blind, Multicenter, Phase 3 Study of in Patients With Advanced NET (Gastrointestinal (GI) or Lung Origin) (RADIANT-4) (NCT01524783)	302	[44]
Sunitinib (SU011248, SUTENT) vs. placebo	All grades 59% vs. 39%	Phase 3, Randomized, Double-Blind Study In Patients With Progressive Advanced/Metastatic Well-Differentiated Pancreatic Islet Cell Tumors (NCT00428597)	171	[45]
Octreotide LAR (long-acting release) 30 mg vs. placebo	All grades 22% vs. 28%	Placebo-controlled, double-blind, prospective, randomized Phase 3 Study on Antiproliferative Effect of Octreotide in Patients With Metastasized Neuroendocrine Tumors of the Midgut (NCT00171873)	85	[46]
Pazopanib (GW786034)	All grades 63%	Open Label Phase 2 Study in Advanced Low-Grade or Intermediate-Grade Neuroendocrine Carcinoma	52	[47]

**Table 5 jcm-09-02468-t005:** Diarrhea outcome associated with carcinoid syndrome upon treatment with somatostatin analogs (SSAs) and tryptophan hydroxylase inhibitors.

Treatment	Diarrhea Outcome	Other Outcomes	Type of Study	References
Octreotide (150 μg three times daily)	Rapid palliation in 88% of patients	Reduction in 5-HIAA levels in 72% of patients	Single arm trial	[60]
Intramuscular octreotide LAR (10, 20, or 30 mg every 4 weeks) + subcutaneous octreotide (every 8 h)		Best control of flushing with 20 mg and subcutaneous treatments	Randomized phase 3 trial	[61]
Octreotide LAR depot 30 mg, 20 m, and 40 mg for at least 4 months	Diarrhea improvement in 48% of patients	Flushing improvement in 60% of patients	Retrospective study	[62]
Octreotide LAR (doses equal or higher than 30 mg every 4 weeks)	Diarrhea improvement in 62% of patients	Flushing improvement in 56% of patients	Retrospective chart-review	[63]
Lanreotide depot vs. placebo	Reductions of days patients reported symptoms: 16 weeks and sustained over 32 weeks	Reductions of days patients reported symptoms of flushing: 16 weeks and sustained over 32 weeks	Double blind, randomized placebo controlled phase 3 clinical trial	[64]
Octreotide 200µg 2/3 times daily for 1 month followed by lanreotide 30 mg every 10 days for 1 month and vice versa	Disappearance or improvement of diarrhea in 45.4% of patients on lanreotide and 50% of patients on octreotide	Disappearance or improvement in flushes in 53.8% of patients on lanreotide and in 68% on octreotide; reduced urinary 5-HIAA levels with both treatments	Prospective, open, multicenter, crossover study	[65]
Octreotide LAR above 30, 40 or 60 mg every 4 weeks	Improvement or resolution of diarrhea in 79% of patients after dose escalation	Improvement or resolution of flushing in 81% of patients after dose escalation	Multicenter retrospective chart review study	[66]
Octreotide LAR > 30 mg	Decrease in diarrhea in 62% of patients	Decrease in flushing in 91% of patients; Reduction in 5-HIAA levels in 23% of the patients	Retrospective review	[67]
Pasireotide LAR (60 mg) vs. octreotide LAR (40 mg)	Diarrhea (all grades) 6% with pasireotide LAR Diarrhea (all grades) 2% with octreotide LAR	42% of flushing events after 6 months with pasireotide LAR 49% of flushing episodes with octreotide LAR	Randomized, double-blind, phase 3 study	[68]
Lanreotide 120 mg vs. Placebo every 4 weeks, followed by 32 weeks’ initial open label lanreotide	Days with moderate/severe diarrhea and/or flushing with lanreotide 23.4% vs. placebo 35.8%; diarrhea scores improved between double-blind and initial open label treatment		Randomized, placebo-controlled, double-blind and 32 week open-label study	[59]
Lanreotide Autogel^®^ (120 mg every 14 days)	−	−	Open label, phase 2	NCT02651987 [69]
High doses of SSAs		Adverse events 15%	Retrospective analysis	[58]
High doses of octreotide LAR		Uncommon treatment discontinuations due to adverse events; cholelithiasis, may increase with longer duration of treatment	Review	[57]
Telotristat etiprate 250 mg or 500 mg three times/day +SSAs vs. placebo + SSAs		≥30% bowel movement frequency reduction in ≥50% of patients (20%, 44%, and 42% for placebo, telotristat 250 mg and 500 mg, respectively); reduced mean urinary 5HIAA (both treatments) vs. placebo at week 12 (*p* < 0.001)	Randomized, placebo-controlled, parallel group, multicenter, double-blind, phase 3 Study (TELESTAR)	[70]
Telotristat etiprate 250 mg or 500 mg three times/daily + SSAs vs. placebo + SSAs	19%, 16% and 8% diarrhea for placebo, telotristat 250 mg and 500 mg, respectively	Reductions in 5-HIAA (−54.0% and −89.7% median difference from placebo for telotristat 250 mg and 500 mg, respectively)	Randomized, placebo-controlled, multicenter, double-blind, phase 3 Study (TELECAST)	[71]

5-HIAA: 5-hydroxyindoleacetic acid, LAR: long-acting release.

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
