# Peer review of "Differential Diagnosis and Management of Diarrhea in Patients with Neuroendocrine Tumors"

_jcm, 2020, doi:10.3390/jcm9082468_

Round 1

Reviewer 1 Report

The authors have largely revised the main causes of diarrhea, including the diagnosis the secretory diarrhea associate to some functioning neuroendocrine tumors. While the paper has some strengths, it seems a little confusing because the loss of the focus in NETs. Especially in section 4 where the authors revise the main causes of diarrhea not related with NETs, as non-functioning NETs, by definition, do not associate diarrhea, being this an iatrogenic consequence of their treatment. Indeed, in table 3 and 5, NETs do not appear as a cause or as a mechanism of PEI. This section should be reduced, eliminated or focused differently

SPECIFIC ASPECTS

Table 1 and table 2 are redundant. Table 2 should be eliminated although some of its content can be incorporated in table 1.

In table 1 (and 2) the row corresponding to PPoma should be eliminate. In addition,  the authors have moved the information provided by Braga et al,  and as these authors clearly remark, PPomas are non-functioning tumors which only, very unusually, associate diarrhea. Therefore, PPoma cannot be held responsible of the watery diarrhea syndrome, a term associate to the Wener-Morrisonn syndrome.

In fig. 1 The WDS related box should be eliminated or related to the Werner-Morrison syndrome

Section 4 should be merged with section 6.

Author Response

Reviewer 1

The authors have largely revised the main causes of diarrhea, including the diagnosis the secretory diarrhea associate to some functioning neuroendocrine tumors. While the paper has some strengths, it seems a little confusing because the loss of the focus in NETs. Especially in section 4 where the authors revise the main causes of diarrhea not related with NETs, as non-functioning NETs, by definition, do not associate diarrhea, being this an iatrogenic consequence of their treatment. Indeed, in table 3 and 5, NETs do not appear as a cause or as a mechanism of PEI. This section should be reduced, eliminated or focused differently

Thanks for your interest in our manuscript. Actually, we preferred to comprehensively consider the causes of diarrhea in NET patients, and not only those related to functioning NETs, for the sake of completeness. However, we have largely revised the paper following your valuable suggestions and have refocused section #4 merging it with section #6, also according to the comments of the other reviewer.

SPECIFIC ASPECTS

Table 1 and table 2 are redundant. Table 2 should be eliminated although some of its content can be incorporated in table 1.

Thanks for your suggestion. We have now removed Table 2 and incorporated some of its contents into Table 1.

In table 1 (and 2) the row corresponding to PPoma should be eliminate. In addition,  the authors have moved the information provided by Braga et al,  and as these authors clearly remark, PPomas are non-functioning tumors which only, very unusually, associate diarrhea. Therefore, PPoma cannot be held responsible of the watery diarrhea syndrome, a term associate to the Wener-Morrisonn syndrome.

Thanks for your major remark. We have now removed PPomas from the manuscript in line with your suggestion and at the light of their rarity.

In fig. 1 The WDS related box should be eliminated or related to the Werner-Morrison syndrome.

We have now revised Figure 1 in line with your suggestion.

Section 4 should be merged with section 6.

We have merged Section #4 with Section #6 in line with your suggestion.

Reviewer 2 Report

The authors present a review of the contemporary literature with the aims to establish a practical guidance for the differential diagnosis and management of diarrhea in patients with GEP-NENs. In my view, the manuscript is of interest to the clinical community of NET specialists and summarizes well valuable information; therefore, I believe it would be of interest to the readership of JCM. However, there are some major points that need to be addressed to improve the quality of the manuscript.

  1. Please provide a PRISMA flow diagram and the search strategy used to identify eligible studies from relevant databases (PubMEd, EMBASE, Cochrane Central, WOS?) in order to comply with JCM's guidelines on the conduct of a systematic Review.
  2.  Although the title of the present review claims that its focus is centered in the differential diagnosis of diarrhea and also the diarrhea management in the setting of patients with NEN, the part of differential diagnosis is neither clear nor comprehensive, probably due to a lack of data. It would be therefore more appropriate to make some general recommendations, as in Fig 1, but focus in the pathophysiology of diarrhea in GEP-NEN patiens and adjust the title and the manus accordingly.
  3.  The mechanisms underlying diarrhea in NENs are well presented. However, in the subset of small intestinal NENs, there are more complex mechanisms apart from the ones described in the manuscript, including the development of mesenteric fibrosis that can contribute to profound diarrhea by partial obstruction-knicking of the bowel but also venous stasis when the mesenteric vessels are engaged and compressed in the root of the mesentery, leading to malabsorption. In addition, in cases of mesenteric fibrosis with retroperitoneal extension, the tumor secretory products may exceed the detoxifying capacity of the liver, or bypass it, draining directly into the systemic circulation through retroperitoneal lymphatic spread, causing this way profound CS refractory to conventional medical treatment (Ref: Koumarianou et al. J. Clin. Med. 2020, 9(6), 1777; https://doi.org/10.3390/jcm9061777)
  4.  The introduction part should focus more specifically in NEN-related diarrhea and shortened accordingly. For example, the 2nd paragraph of page 2(lines 53-59) on NEN grading could be ommited or shortened. The same applies to paragraphs of etiology criteria that are too lengthy and not directly related to NEN disease (2.1-21.1-2.1.2-2.1.3-2.14; page 2 and 3, lines 84-127).
  5.  Please rephrase-correct the sentence in page 5, lines168-170, as it is not true that 25% of NETs are associated with ZES.
  6.  Please present the figures of diarrhea, as an adverse effect of medical treatments used in GEP-NENs from the pertinent RCTs on GEP-NEN management, specifically for SSAs, mTOR inhibitors, TKIs etc in a comprehensive Table.
  7.  Section 6 on Differential diagnosis although it is a good concept, it is too general and the Authors do not provide practical tests or an easy to follow algorithm to guide clinicians. In my view, this section could also be shortened.
  8.  Please provide a comprehensive figure with a practical algorithm used at the Authors institution with medications targeting diarrhea in GEP-NENs. 
  9.  Paragraphs on SSAs and telotristat are lengthy and difficult to follow. Please consider adding a table with data from the included studies on control of CS and mainly diarrhea.
  10.  Please refer to latest ESMO guidelines on IF-a in NENs, e.g. in SSR negative tumors (page 14, lines533-534).
  11. Conclusions rather than an expert opinion should be presented in a shorter version in the end of the manuscript.

Author Response

Reviewer 2

The authors present a review of the contemporary literature with the aims to establish a practical guidance for the differential diagnosis and management of diarrhea in patients with GEP-NENs. In my view, the manuscript is of interest to the clinical community of NET specialists and summarizes well valuable information; therefore, I believe it would be of interest to the readership of JCM. However, there are some major points that need to be addressed to improve the quality of the manuscript.

Thanks for your interest in our manuscript, which has now largely been revised in line with your valuable suggestions and those of the other reviewer.

  1. Please provide a PRISMA flow diagram and the search strategy used to identify eligible studies from relevant databases (PubMEd, EMBASE, Cochrane Central, WOS?) in order to comply with JCM's guidelines on the conduct of a systematic Review.

Thanks for this major suggestion. However, ours is not a systematic review, but rather a narrative review. We have clarified this in the text.

  1. Although the title of the present review claims that its focus is centered in the differential diagnosis of diarrhea and also the diarrhea management in the setting of patients with NEN, the part of differential diagnosis is neither clear nor comprehensive, probably due to a lack of data. It would be therefore more appropriate to make some general recommendations, as in Fig 1, but focus in the pathophysiology of diarrhea in GEP-NEN patiens and adjust the title and the manus accordingly.

Actually, we preferred to comprehensively consider the causes of diarrhea in NET patients, and not only those related to functioning NETs, for the sake of completeness. However, we have largely revised the paper following your valuable suggestions and have extensively revised the section on other causes of diarrhea and differential diagnosis.

  1. The mechanisms underlying diarrhea in NENs are well presented. However, in the subset of small intestinal NENs, there are more complex mechanisms apart from the ones described in the manuscript, including the development of mesenteric fibrosis that can contribute to profound diarrhea bypartial obstruction-knicking of the bowel but also venous stasis when the mesenteric vessels are engaged and compressed in the root of the mesentery, leading to malabsorption. In addition, in cases of mesenteric fibrosis with retroperitoneal extension, the tumor secretory products may exceed the detoxifying capacity of the liver, or bypass it, draining directly into the systemic circulation through retroperitoneal lymphatic spread, causing this way profound CS refractory to conventional medical treatment (Ref: Koumarianou et al.  Clin. Med. 20209(6), 1777; https://doi.org/10.3390/jcm9061777).

We have now better clarified the mechanisms of diarrhea in small intestinal NENs with proper reference to the paper you suggested.

  1. The introduction part should focus more specifically in NEN-related diarrhea and shortened accordingly. For example, the 2nd paragraph of page 2(lines 53-59) on NEN grading could be ommited or shortened. The same applies to paragraphs of etiology criteria that are too lengthy and not directly related to NEN disease (2.1-21.1-2.1.2-2.1.3-2.14; page 2 and 3, lines 84-127).

Thanks for your suggestion. We have reduced these introductory parts accordingly in order to better focus the manuscript.

  1. Please rephrase-correct the sentence in page 5, lines168-170, as it is not true that 25% of NETs are associated with ZES.

We have removed this sentence for the sake of brevity; information on the prevalence of ZES in NETs is reported also earlier in the paragraph.

  1. Please present the figures of diarrhea, as an adverse effect of medical treatments used in GEP-NENs from the pertinent RCTs on GEP-NEN management, specifically for SSAs, mTOR inhibitors, TKIs etc in a comprehensive Table.

Thanks for this major suggestion. We have included this table as per your request.

  1. Section 6 on Differential diagnosis although it is a good concept, it is too general and the Authors do not provide practical tests or an easy to follow algorithm to guide clinicians. In my view, this section could also be shortened.

We agree with your suggestion, also in line with the comments by the other reviewer. We have now incorporated section #6 into section #4, and extensively shortened the entire manuscript.

  1. Please provide a comprehensive figure with a practical algorithm used at the Authors institution with medications targeting diarrhea in GEP-NENs. 

We have included a practical algorithm for the treatment of diarrhea in GEP-NETs as per your suggestion.

  1. Paragraphs on SSAs and telotristat are lengthy and difficult to follow. Please consider adding a table with data from the included studies on control of CS and mainly diarrhea.

We have shortened those paragraphs and included a dedicated table.

  1. Please refer to latest ESMO guidelines on IF-a in NENs, e.g. in SSR negative tumors (page 14, lines533-534).

We have now updated references accordingly.

  1. Conclusions rather than an expert opinion should be presented in a shorter version in the end of the manuscript.

The paragraph has been extensively shortened and the heading has been changed to “Conclusions”.

Round 2

Reviewer 1 Report

The authors have answered properly my queries

Reviewer 2 Report

The paper' quality is now significantly improved, following the Author's extensive revision.

This manuscript is a resubmission of an earlier submission. The following is a list of the peer review reports and author responses from that submission.